# Feature Reviews of the Molecular Mechanisms of Nasopharyngeal Carcinoma

**DOI:** 10.3390/biomedicines11061528

**Published:** 2023-05-25

**Authors:** Li-Jen Liao, Wan-Lun Hsu, Chi-Ju Chen, Yen-Ling Chiu

**Affiliations:** 1Department of Otolaryngology Head and Neck Surgery, Far Eastern Memorial Hospital, New Taipei City 220, Taiwan; deniro@mail2000.com.tw; 2Department of Electrical Engineering, Yuan Ze University, Taoyuan 320, Taiwan; 3Master Program of Big Data Analysis in Biomedicine, College of Medicine, Fu Jen Catholic University, New Taipei City 242, Taiwan; wanlun0112@gmail.com; 4Data Science Center, College of Medicine, Fu Jen Catholic University, New Taipei City 242, Taiwan; 5Institute of Microbiology and Immunology, National Yang Ming Chiao Tung University, Taipei 112, Taiwan; 6Department of Medical Research, Far Eastern Memorial Hospital, New Taipei City 220, Taiwan; 7Graduate Institute of Medicine and Graduate Program in Biomedical Informatics, Yuan Ze University, Taoyuan 320, Taiwan

**Keywords:** molecular mechanisms, nasopharyngeal carcinoma, Epstein–Barr virus

## Abstract

Nasopharyngeal carcinoma (NPC) is rare in most parts of the world but endemic in southern Asia. Here, we describe the molecular abnormalities in NPC and point out potential molecular mechanisms for future therapy. This article provides a brief up-to-date review focusing on the molecular pathways of NPC, which may improve our knowledge of this disease, and we also highlight some issues for further research. In brief, some heritable genes are related to NPC; therefore, people with a family history of NPC have an increased risk of this disease. Carcinogenic substances and Epstein–Barr virus (EBV) exposure both contribute to tumorigenesis through the accumulation of multiple genomic changes. In recent years, salted fish intake has decreased the impact on NPC, which implies that changing exposure to carcinogens can modify the risk of NPC. Eradication of cancer-associated viruses potentially eradicates cancer, and EBV vaccines might also prevent this disease in the future. Screening patients by using an EBV antibody is feasible in the high-risk group; plasma EBV DNA measurement could also be conducted for screening, prognosis, and monitoring of this disease. Understanding the molecular mechanisms of NPC can further provide novel information for health promotion, disease screening, and precision cancer treatment.

## 1. Introduction

Nasopharyngeal carcinoma (NPC) is rare in most parts of the world but is endemic in southern Asia [1]. According to histological features, including degrees of keratinization and differentiation of the tumor cells, NPC can be classified into keratinizing type (WHO type I), non-keratinizing differentiated type (WHO type II), non-keratinizing undifferentiated (WHO type III), and basaloid squamous-cell carcinoma. Nonkeratinizing NPC is the most common subtype, accounting for >95% of cases in endemic areas. The tumorigenesis of NPC is a sequential multistep process that transforms normal nasopharyngeal epithelium into cancer cells [2] (Figure 1). The key events include carcinogen exposure, mutations in genes regulating immune responses, epigenetic processes, and signal transduction, as well as the genome instability promoted by Epstein–Barr virus (EBV). Here, we provide a brief up-to-date review focusing on the molecular mechanism governing NPC tumorigenesis and highlight some yet-to-be addressed issues for future research.

## 2. Environmental Risk Factors

The association between dietary factors and NPC has been supported by numerous studies, particularly in high-risk populations such as southern China and southeast Asia. The risk factors include frequent consumption of salt-preserved fish, meats, and eggs [3,4]. The high levels of N-nitrosamines and other carcinogenic compounds found in preserved foods may contribute to the development of NPC [5,6]. However, a recent large population-based case–control study conducted in southern China found that salty fish and preserved food intakes are only weakly associated with NPC risk in adults [7], suggesting that other genetic factors and EBV infection are more significant risk factors for developing NPC. Dietary patterns may also affect NPC risk. Diets high in saturated fat and carbohydrates and low in polyunsaturated fatty acids and flavonoids are associated with greater NPC risk because of their high inflammatory potential [8,9]. On the other hand, the consumption of fresh fruits and vegetables decreases NPC risk [10,11]. Intakes of traditional herbal medicines, tea, and alcohol have all been investigated for their association with NPC; however, the results were inconsistent among the studies [5,12].

Other nondietary factors include Epstein–Barr virus infection, tobacco smoke, occupational exposure to fumes, smoke, dust, and chemicals, including formaldehyde. These carcinogenic materials can induce DNA damage, genome instability, and a dysregulated cell cycle that increase the tumorigenesis of NPC. Exposure to carcinogens can also reactivate latent EBV virus [5,12]. We listed the possible risk factors and degree of their association with NPC in Table 1.

## 3. Implication of Epstein–Barr Virus Infection in NPC

Epstein–Barr virus (EBV) is a ubiquitous human g-herpesvirus infecting more than 90% of the adult human population and that establishes life-long latency in epithelial cells and B cells. Its infection is associated with various lymphoid malignancies, including Burkitt lymphoma, Hodgkin lymphoma, B-cell lymphoma, and NK/T-cell lymphoma, as well as two types of epithelial cancer, nasopharyngeal carcinoma (NPC) and subtypes of gastric carcinoma. EBV has two distinct life cycles, namely, lytic (productive) and latent (persistent). After initial replication in nasopharyngeal epithelial cells, EBV passes through the epithelial cells lining the nasopharynx and infects and replicates in naïve B cells. It eventually establishes a benign and stable latency in infected cells in a state of histone-associated episome [13]. Periodic reactivation of latent virus is required for cell-to-cell spreading to maintain a stable reservoir of latent virus and EBV oncogenesis [14]. The prevalence of EBV is 100% in type 2 and type 3 NPC, and a high level of reactivation is a risk factor for EBV-related NPC [15]. The combination of EBV infection with environmental and genetic factors underlies NPC pathogenesis. New evidence indicates that abortive lytic infection facilitates latency establishment during primary infection and with EBV-associated tumors [16]. Understanding how EBV viral products contribute to NPC tumorigenesis can help us better design EBV-targeting therapies.

During latency, viral gene expression is limited to a small subset of viral genes that are important for genome maintenance and regulatory functions. There are four types of latency that are defined by the expression profile of late genes. EBV infection in NPC is typically represented by a type 2 latency (default program), in which EBV nuclear antigen 1 (EBNA1), latent membrane protein 1 (LMP1), and LMP2 proteins are produced. In addition, noncoding RNAs, such as EBV-encoded RNAs (EBERs), BamH1 A rightward transcripts (BARTs), and BART miRNAs, are expressed. Expressed viral genes provide necessary replication and survival signals to both EBV and host cells. Latent genes seem to be the key drivers in EBV-associated NPC; however, EBV lytic phase proteins, such as BGLF5 (DNase) and BALF3, promote host genome instability and play important roles in NPC tumorigenesis [17,18]. Table 2 summarizes viral proteins/noncoding RNAs found in NPC and their functions.

Similar to any other type of cancer, early NPC detection is key for a good prognosis. Nasopharyngoscopy is the tool of choice to exam nasopharynx and guide biopsy. Narrow-band imaging (NBI) could be applied for the diagnostic screening and accuracy rate increment of a high-risk population of NPC [19]. The percentage of early detection of NPC cases can vary depending on various factors, such as the geographic region, access to health care, and screening practices. However, due to the anatomic location of the nasopharynx and unapparent and nonspecific early symptoms, 75% of NPC is detected in advanced stages (III/IV) in general. Late-stage NPC is associated with a poorer prognosis, as the cancer is more difficult to treat and is more likely to recur. Therefore, it is important to use effective screening methods in endemic areas or for individuals with a family history of NPC. Many studies have shown that measuring serum levels of EBV antibodies and viral DNA can be useful for the diagnosis and prognosis of NPC [15,20,21].

For example, an early study showed that measurements of blood EBV DNA could be a predictor of disease recurrence and overall survival in NPC patients [22]. Later, another study analyzed the plasma levels of EBV DNA, as well as antibodies to viral capsid antigen (VCA) and early antigen (EA), in 111 patients with advanced NPC. The authors also found that high levels of plasma EBV DNA were associated with a poor prognosis and higher risk of death in NPC patients [15]. A recent meta-analysis, which included 47 studies (up to January 2019) with 8382 NPC patients (NPC group) and 15,089 individuals without NPC (control group) using Chinese data libraries, concluded that EBV-DNA, VCA-IgA, EBNA1-IgA, and Rta-IgG were useful in early diagnosis, with EBV-DNA having higher diagnostic accuracy [23]. On the other hand, due to the low prevalence, even EBV antibodies or EBV DNA have high diagnostic sensitivity and specificity [23], and they are not suitable to screen NPC in the general public.

EBV gene products often provide a cell proliferation advantage and anti-death signal. Therefore, targeting the EBV genome or the signaling pathways it stimulates is an attractive way to treat EBV-positive cancers. There are two different approaches used for targeting EBV-positive cancer cells: (1) targeting EBV latent genes and (2) lytic induction therapy. For targeting EBV latent proteins, EBNA1, LMP1, and LMP2 are potential therapeutic targets. Among them, disrupting the function of EBNA1, a protein essential for latent EBV genome replication and maintenance, is gaining the most attention. Strategies include using small molecules or DNA containing the EBNA1 binding sequence or gRNA targeting oriP to interfere with EBNA1 dimerization/oligomerization and DNA binding. The major challenge of these approaches is how to deliver the aforementioned molecules to cancer cells efficiently [27,28]. Moreover, advances have been made through the development of new theranostic agents targeting EBNA1 [29]. Relatively few normal cells contain EBV, whereas the majority of EBV-associated cancer cells contain EBV. Therefore, it is an attractive idea to reactivate EBV as part of an oncolytic treatment [27,30], although it is at an early developmental stage. The lytic induction approach involves administering two types of drugs: the first drug induces lytic gene expression, and the second drug causes cytotoxicity in latently infected tumor cells. Current lytic-induction strategies take advantage of two EBV-encoded kinases, namely, protein kinase encoded by BGLF4 and thymidine kinase encoded by BXLF1, to convert prodrugs such as ganciclovir, acyclovir, and fialuridine into active forms [31,32]. In addition to directly causing cell death, lytic induction increases susceptibility to antiviral agents and may also increase immune responses targeting EBV-positive cells. Early studies using this approach to treat patients with advanced EBV-positive lymphoma and NPC showed promising results [33,34]. Lytic induction reagents are often cell background dependent, meaning some work better than others in a given cell/tumor type background [35]. Identifying chemotherapeutic agents that better induce lytic gene expression and drug combinations that are not overly toxic is essential for this approach to be successful. Studies investigating the mechanisms, including cellular signaling pathways, transcription factors, and viral proteins, involved in controlling the EBV latent-lytic switch will help us to achieve this goal. Histologically, the NPC is classified into WHO type I, II, and III according to keratinizing and differentiation. Molecular markers, including genetic factors and EBV variants, are potentially useful in subtyping NPC for prognosis and treatment [2].

## 4. Genomic Instability and Mutations

Genome instability refers to an increased tendency for errors to occur during DNA replication or repair, leading to mutations, deletions, or other alterations in the genome. Genome instability can occur naturally as a result of abnormal cellular processes (endogenous), or it can be induced by exposure to various environmental factors (exogenous), such as radiation, certain chemicals, or viruses. There is evidence to suggest that genome instability plays a role in the development of NPC, and several factors have been implicated in promoting genome instability in NPC. For example, EBV is known to cause DNA damage and inhibit DNA repair mechanisms in infected cells, which can increase the likelihood of mutations and chromosomal abnormalities [36]. In addition, exposure to certain environmental factors, such as tobacco smoke, alcohol, formaldehyde, and nitrosamines, has been linked to DNA damage and genome instability, which may contribute to NPC development. Genetic factors such as mutations in tumor suppressor genes or DNA repair genes may also increase susceptibility to NPC by impairing the cell’s ability to maintain genomic stability. Epstein–Barr virus (EBV) is known to cause genome instability by several mechanisms. First, EBV, although a rare event, can insert its own genetic material into the host cell’s genome, which can lead to chromosomal rearrangements and mutations [37,38]. EBV can also induce DNA damage directly by introducing oxidative stress and reactive oxygen species into the host cell, which can cause DNA breaks and other types of DNA damage [39,40]. In addition, EBV can interfere with host cell DNA repair mechanisms, leading to the accumulation of DNA damage and increased genome instability [41]. EBV can also affect the expression of genes involved in DNA repair, cell cycle regulation, and apoptosis [42], which can contribute to genome instability and tumorigenesis [36]. Overall, the mechanisms by which EBV causes genome instability are complex and multifaceted and likely involve a variety of pathways and cellular processes. Understanding these mechanisms is important for developing targeted therapies for EBV-associated NPC.

Gene mutations and epigenetic dysregulation are important for NPC tumorigenesis [43]. The loss of chromosomes 3p and 9p may be an important early event for NPC development. Additional chromosomal abnormalities, including copy number increase/decrease in chromosomes 3p, 9p, 11p, 12p, and 14q, as well as epigenetic changes, such as altered widespread DNA methylation status, were identified in NPC [2,44]. Deletion and/or methylation of a number of tumor suppressors, including Ras association domain family member 1 (*RASSF1A*) [45], ADAM metallopeptidase with thrombospondin type 1 motif 9 (*ADAMTS9*), protein tyrosine phosphatase, receptor type G (*PTPRG*) [46], and several other genes have been mapped to the chromosome 3p21.3 region [47,48,49,50,51], as well as p16 on chromosome 9p [52]. NPC has skewed ethnic and geographic distributions; how genetics contribute to the disease mechanism is poorly understood. To better characterize the genetic events in NPC, genome-wide association studies (GWAS), such as exome and genome sequencing, have been carried out by many groups. Lin et al. published the first genome-wide sequence analysis of NPC, in which the mutational landscape of 128 NPC cases was determined using whole-exome, targeted deep sequencing, and SNP array analysis. The study identified 144 genes recurrently mutated in these cases. Mutations in *PIK3CA* and *TP53*, previously found in NPC, were confirmed in the study. Seven new genes, including *BAP1, ERBB2, ERBB3, KRAS, NRAS, KMT2D (MLL2)*, and *TSHZ3*, were identified to be significantly associated with NPC [53].

Based on several other GWAS, the major histocompatibility complex (MHC) region on chromosome 6p21, which contains human leukocyte antigen (HLA) genes, was most strongly associated with NPC [54,55]. HLA types associated with NPC include *HLA-A2*, *HLA-A11*, *HLA-B46*, and *HLA-B58*. These associations suggest that the immune response plays an important role in the development and progression of NPC. It is worth noting that the association of HLA types with NPC may vary depending on the geographic region and ethnic group. For example, *HLA-A2* association with NPC is common in Asians [56,57]. Caucasians with *HLA-A2* in the US have a significantly lower risk than those with other antigens at the A locus [58]. A study comparing single nucleotide polymorphisms (SNPs) of 1583 NPC cases to 1894 controls of southern Chinese descent not only confirmed the association but also identified three additional susceptibility loci: *TNFRSF19* on 13q12, *MDS1-EVI1* on 3q26, and the *CDKN2A-CDKN2B* gene cluster on 9p21 [59]. The loss of heterozygosity (LOH) of chromosomes 3p and 9p in NPC has also been linked to EBV infection and genomic instability [60,61]. Polymorphisms of metabolic genes that have major functions in the metabolism of carcinogens, such as cytochrome P450 (CYP) genes, were linked to the risk of NPC. While no association between polymorphism of *CYP2A13* and NPC susceptibility was found in a Taiwan cohort [62] and Cantonese population [63], polymorphism of *CYP2E1* was found to be linked to NPC risk in several studies, including a case–control study conducted in Taiwan [64,65,66,67,68]. The links of several CYP genes to NPC suggest that the interaction between carcinogens and polymorphisms in metabolic genes awaits further investigation. Epigenetic changes also contribute to the dysregulation of various signaling cascades in NPC. Hypermethylation was the most frequently reported in cancer-associated genes and led to disrupted DNA repair, stress response, and apoptotic signaling in NPC. EBV-encoded LMPs may induce hypermethylation of some tumor suppressor genes and facilitate maintenance of viral latency, cell proliferation, and tumor survival [69] (Table 3).

## 5. Frequently Altered Oncogenic Pathways

It is likely that genetic aberrations, environmental factors, and EBV infection cooperate in NPC tumorigenesis, during which cellular signaling pathways are often altered. Clarifying how these oncogenic pathways contribute to NPC can potentially help us find new druggable targets [70]. Based on GWAS, mutations in NPC can be categorized into the following pathways:

### 5.1. NF-κB Signaling

Dysregulation of NF-κB, which is important for angiogenic and tumorigenic regulation, is often observed in cancer. The constitutive activation of NF-κB is associated with the dysregulation of the transcription of genes that encode cytokines, chemokines, adhesion factors, and inhibitors of apoptosis [71]. By EMSA, Western blotting, and immunohistochemical staining, constitutive activation of NF-κB complexes, either p50/p50/Bcl3 or p50/RelB, was found in almost all EBV-positive NPC tumors and cell lines [72]. More recently, GWAS have shown that dysregulation of the NF-κB signaling pathway is caused by various somatic alterations in genes encoding members of this pathway [73,74]. Multiple truncating mutations in three negative regulators, including *NF-κB inhibitor alpha* (*NFKBIA*), *cylindromatosis* (*CYLD*), and TNF alpha-induced protein 3 (TNFAIP3), were found in NPC primary tumors in a whole-exome study at a high frequency [70,73]. *NFKBIA* regulates NF-κB nuclear translocation. *CYLD* encodes deubiquitinating enzymes (DUBs), which regulate cell growth and invasiveness by untying K63-linked ubiquitin chains through deubiquitination [75]. *TNFAIP3*, the gene encoding A20, is associated with a wide panel of inflammatory pathologies [76]. Mutations in these negative regulators can lead to constitutive activation of the NF-κB pathway in NPC. It has also been demonstrated that in the majority of NPCs examined, EBV-encoded LMP1 expression underpins constitutive NF-κB activation, which contributes to inflammatory and immune escape [74].

### 5.2. PI3K/AKT/mTOR Signaling Pathway

The *PI3K/AKT* pathway regulates a number of cellular processes representing cancer hallmarks, such as cell growth, proliferation, anti-apoptosis, migration, and angiogenesis [77]. The serine/threonine kinase AKT is a central node in the cell signaling cascade downstream of growth factors, cytokines, and other cellular stimuli by generating the second messenger phosphatidylinositol 3,4,5-trisphosphate. Dysregulation in *PI3K/AKT*, either altered expression or mutation of components of the pathway, is found in many types of cancer [78,79]. The pathway also plays a critical role in the tumorigenesis of NPC. Several studies have reported genetic alterations in this pathway in NPC, including mutations in *PIK3CA* (encoding the p110α subunit of *PI3K*), *PTEN* (a negative regulator of the pathway), and *AKT1/2/3* (encoding AKT isoforms). These mutations result in increased activity of the *PI3K/AKT* pathway, contributing to the development and progression of NPC. Mutations in *PIK3CA* promote cell proliferation and migration and inhibit apoptosis in NPC cells [80]. In an NPC GWAS, genes involved in the *ErbB-PI3K* signaling pathway were mutated in 30/128 cases [53]. *PIK3CA* mutation was one of the most frequently dysregulated oncogenes, activated by hot-spot mutations (for example, those encoding His1047Arg and Glu545Lys) and amplification. Mutations in *ERBB2*, *ERBB3*, and *KRAS* were also found. In a separate study [81], mutations in *PI3K/AKT* pathway activators or regulators (*PIK3CA, PTEN, ERBB3, BRAF1, NF1, FGFR2, FGFR3*) were observed, although less frequently. Together, these findings suggest that *ERBB-PI3K* pathway activation by genetic alterations exacerbates NPC malignancy [53]. In addition to genetic alterations, the *PI3K/AKT* pathway can be activated by extracellular signals such as growth factors and cytokines, which bind to their receptors on the cell surface and trigger downstream signaling cascades. LMP1 directly activates *PI3K*, leading to Akt phosphorylation and activation of several downstream signaling pathways [81]. The activation of the *PI3K/Akt* pathway by EBV is thought to contribute to the development of NPC.

### 5.3. TP53 and Cell Cycle Regulators

A genomic landscape study revealed that the G1/S cell cycle transition was affected in 28% of NPCs, with most of the alterations occurring in *TP53* and *CDKN2A* [53]. Mutations in *TP53*, encoding the p53 tumor suppressor, regulate the cell cycle. A relatively low incidence of *TP53* mutations occurs in NPC compared to the incidence rate (85%) found in head and neck cancers [82]. However, *TP53* mutations are consistently detected more frequently in recurrent and metastatic NPC and are associated with a significant reduction in disease-free and overall survival [2,81,83]. This is likely through its role in cell cycle regulation rather than genomic instability caused by defective p53-dependent DNA repair machinery [74].

Genes controlling the G1/S transition are frequently mutated in NPC. Early driver events in NPC likely involve dysregulation of the cell cycle. Homozygous deletions *CDKN2A* (p16) and *CDKN2B* (p15), two negative G1/S regulators, were found in NPC primary tumors [84]. More recent GWAS showed that the most frequent genome deletion occurs in the 9p21 region spanning *CDKN2A* and *CDKN2B* [53]. Amplification at the *CCND1* (cyclin D1) [53,85] and *MYC* [53,86] loci may exacerbate cell cycle dysregulation in NPC.

The tumor suppressor gene *RASSF1A* regulates mitosis and maintains genomic stability by repressing cell cycle-related protein *cell division cycle 20* (*CDC20*) and microtubule proteins. Loss of *RASSF1A* expression in NPC, either due to 3p21.3 deletion or promoter hypermethylation, is associated with NPC genome instability and tumorigenesis [45,87].

### 5.4. TGF-β/SMAD

TGF-β/SMAD has a biphasic action in tumorigenesis: a tumor suppressor at the early stage and a tumor promoter at the late stage, suggesting that its function is context dependent. TGF-β signaling regulates G1/S cell cycle transition, apoptosis, senescence, epithelial-to-mesenchymal transition (EMT), metastasis, angiogenesis, and immune responses [88]. Altered TGF-β/SMAD signaling pathways have been found to be frequent in NPC [89,90]. In a GWAS, a 24.3% (17/70 tumors) rate of TGF-β/SMAD pathway gene alteration was observed in NPC, targeting *TGFBR2*, *TGFBR3*, *ACVR2A*, and *SMAD4* [74].

Downregulation of *TGFBR2*, mutations in which attenuated canonical TGF-β-mediated SMAD signaling, was often found in clinical tumor samples when compared with adjacent histological normal epithelial cells and infiltrating lymphocytes [74]. In another study, 409 cancer-related genes were deep sequenced for 33 NPC paired tumor/normal cells. The authors found that three samples contained *TGFBR2* missense mutations and one contained *SMAD4* missense mutation. In addition, patients harboring mutations in TGF-β/SMAD signaling were associated with poor overall survival and poor recurrence-free survival. All these mutations in *TGFBR2* and *SMAD4* abrogate SMAD-dependent TGF-β signaling, providing evidence that dysregulated TGF-β signaling contributes to exacerbating NPC pathogenesis [89]. Together, these results indicate that dysregulation of TGF-β/SMAD signaling may play an important role in NPC tumorigenesis.

### 5.5. Wnt/β-Catenin Pathway

The Wnt/β-catenin pathway is involved in the development of embryo and adult tissue hemostasis. It contributes to the development and progression of some solid and hematologic malignancies [91]. A recent pooled meta-analysis of 1179 patients with NPC indicated that elevated β-catenin expression was associated with a poor overall survival rate [92]. Recently, N7-methylguanosine tRNA modification, one of the most abundant modifications in tRNAs, was reported to promote tumorigenesis and chemoresistance through the Wnt/β-catenin pathway in NPC [93]. LMP1 regulates host cell proliferation, apoptosis, transformation, metastasis, and immune evasion via the NF-κB pathway. In addition to the NF-κB pathway, it can promote the accumulation of β-catenin and lead to dysplasia via the Wnt/β-catenin pathway, which contributes to the development and progression of solid malignancies [91].

### 5.6. Epigenetic Regulation

Epigenetic dysregulation is a hallmark of NPC tumorigenesis. Host genetics, EBV infection, and environmental factors contribute to epigenetic changes, ultimately leading to NPC development. Epigenetic alterations in NPC include DNA methylation, histone modifications, and noncoding RNA dysregulation [83]. In a methylome analysis, NPC had the highest hypermethylation frequency among ten cancer types. Important DNA methylation changes are frequently found in several critical regions, such as the 3p21.3, 9p21, and 6p21.3 regions. Human leukocyte antigen (HLA), a major genetic determinant for NPC pathogenesis, is located at 6p21.3 [94]. Several tumor suppressors, including *CDKN2A* and *RASSF1*, were in these regions. Downregulation of these promoters contributed to NPC tumorigenesis [88]. Pathway analysis of genomic events from 128 NPC cases revealed that the chromatin-modification pathway was among the most frequently affected pathways. A total of 67 altered genes in 54 cases with NPC affecting multiple chromatin modification processes were identified. *ARID1A* (AT-rich interactive domain-containing protein 1A) was identified to be the most affected gene [53]. *ARID1A* is a key subunit of the SWI/SNF chromatin remodeling complex, which plays a crucial role in regulating gene expression by altering chromatin structure. Downregulation or loss of *ARID1A* expression was found to be a common epigenetic alteration in NPC, which may contribute to NPC tumorigenesis by promoting cell proliferation, invasion, and metastasis [95]. One study found that *ARID1A* mutations were present in 17.5% of NPC cases and were associated with poorer overall survival [95].

EBV infection plays a critical role in epigenetic alteration. EBV-driven global-wild epigenetic silencing includes many tumor suppressor genes. Key epigenetic drivers for (1) DNA methylation (*DNMT1*, *TET1*, *IDH2*), (2) histone modification (KMT2c, EZH2, EED, and BMI1), and chromatin remodeling (*ARID1A*, *BRD7*, *PRDM5*, *EP300*, and *HELLS*) were identified [2].

## 6. Immune Evasion, Immunotyping, and Precise Medication

Nasopharynx-associated lymphoid tissue (NALT) is different from other lymphoid tissue from organogenesis. NALT plays an important role in mucosa immune response [96]. Secretion of EBV specific IgA from NALT to the serum can serve as a tool for NPC screening in high-risk group. New insight into immunity has developed in recent years, and NPC is usually infiltrated with various stromal cells. The programmed death-1/programmed death-ligand 1 (PD-1/PD-L1) pathway has been suggested to play an important role in T-cell tolerance and tumor immune escape. The expression of PD-1 on tumor-infiltrating lymphocytes and PD-L1 on tumor tissue are detected and predict poor outcome [97]. PD-1/PD-L1 co-expression reflected the selective suppression of cytotoxic lymphocytes in the tumor microenvironment and predicted recurrence and metastasis of NPC after conventional therapies. Blocking this pathway in patients with co-expression of PD-1/PD-L1 provides a potential therapeutic target for NPC [97] (Figure 1).

The tumor immune microenvironment (TIME) consists of an immunostimulant TIME and an immunosuppressive TIME. TIME combined with cellular and acellular components. Tumor cells, fibroblasts, endothelial cells, and leukocytes are the main cells of the TIME, while EBV exosomes, cytokines, and chemokines provide mediators between cells and influence tumorigenesis, metastasis, and immune evasion. The microenvironment is heterogeneous and dynamic. The TIME component also serves as a prognostic biomarker and potential target for novel therapy [98,99].

Since NPC expresses the EBV genome, adoptive T-cell therapy has emerged as a potential therapeutic strategy for NPC. Expression of PDL-1 by cancer cells can inhibit the effector function of adoptively transferred EBV-specific T cells. The combination of EBV-specific adoptive T-cell therapy and programmed cell death-1 blockade therapy has been reported to be more successful [100].

## 7. Future Perspectives and Challenges

Remarkable progress has been made in the last decade, and the exact molecular mechanisms of NPC have been more clearly defined. Translation of the results of studies still depends on more solid evidence. Research focusing on the genomic, epigenomic, and immune landscapes of NPC is required. Furthermore, immunotherapy might become an important treatment modality in the future.

## 8. Conclusions

NPC tumorigenesis is a complex process that involves host genetics, viral infection, and environmental factors. It is characterized by a comparatively low mutation rate, extensive hypermethylation, frequent chromosomal abnormalities, and copy number alterations [2]. Primary prevention for NPC includes decreasing carcinogen exposure; decreasing salted fish and preserved food consumption; avoiding cigarette smoking, working dust, and chemical exposure; increasing vegetable consumption; and developing EBV vaccines. The secondary prevention strategy is to identify high-risk populations by screening patients using EBV antibodies and analyzing the viral genome.

## Figures and Tables

**Figure 1 biomedicines-11-01528-f001:**
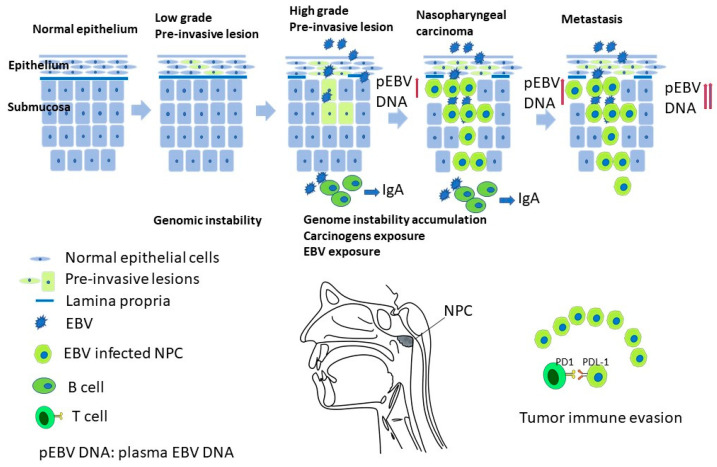
A proposed multiple-step model of NPC carcinogenesis, understanding the mechanism would provide insight into the prevention, screening, subtyping, and precision cancer treatment of NPC.

**Table 1 biomedicines-11-01528-t001:** Summary of possible risk factors for NPC [5,12].

Dietary Factors	Degree of Association
Salt-preserved foods	Moderate to strong
High fat diet/lack of fresh fruits and vegetables	Moderate
Herbal medicines, tea, alcohol	Weak
**Nondietary factors**	
Epstein–Barr virus infection	Strong
HLA polymorphisms	Strong
Tobacco smoke	Weak
Chemicals e.g., formaldehyde, occupational dust/smoke	Weak to moderate increase
Chronic respiratory inflammation	Moderate increase

**Table 2 biomedicines-11-01528-t002:** Summary of the EBV products that are detected in NPC tissue and their main functions [24,25,26].

EBV Associated Substances	Molecular Mechanisms
**Latent EBV proteins**	
LMP1	Invasion and metastasis Activates oncogenic signaling pathways
	Anti-apoptosis
	Inhibits differentiation of squamous cells
	Metabolic reprogramming
	Immune evasion
LMP2	Promotes cell survival
	Maintains cell stemness
	Metabolic reprogramming
EBNA1	EBV episome maintenance
	Disrupts PML nuclear bodies
	Promotion of angiogenesis
	Evasion from immune surveillance
**Lytic EBV proteins**	
BZLF1 (Zata)	Activator of lytic gene/upregulates IL-10
BRLF1 (Rta)	Activator of lytic gene/promotes paracrine of MMP9
BMRF1 (EA-D)	DNA polymerase processivity factor
BGLF5	DNase/genome instability
BALF3	Genome instability
**Noncoding RNA**	
EBER	Anti-apoptosis
	Enhances TNF- α levels
miRNAs	Blocks expression of host RNAs
miR-BART	Potentiates tumor growth
	Immune evasion
	Anti-apoptosis

Abbreviation: PML, promyelocytic leukemia.

**Table 3 biomedicines-11-01528-t003:** Summary of reported genomic changes in NPC [36].

Location of Chromosome	Genes Abnormalities	Tumor Associated Function
Chr.3p21.3	*RASSF1*	Cell growth, proliferation
Chr.3q26.3	*PIK3CA*	Oncogene
Chr.6q	*HLA-A/B/C*	Immune regulation
Chr.7	*MAD1L1*	Mitotic checkpoint
		gene/chromosomal instability
Ch.11q13.3	*Cyclin D1 (CCND1)*	Cell cycle regulatory protein
Chr.12p13.3	*Lymphotoxin-beta receptor (LTBR) gene*	Tumor necrosis factor receptor (TNFR) family, activate NF-κB and c-Jun N-terminal kinase
Chr.14q13.2	*NFKBIA*	Inflammatory pathway
Chr.15q22-q24	*CYP1A1*	Detoxification
Chr.16q12.1	*CYLD*	Promote apoptosis
Chr.17p13.1	*TP53*	Tumor suppression/cell cycle

## Data Availability

This study did not report any data.

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
