# Peer review of "Feature Reviews of the Molecular Mechanisms of Nasopharyngeal Carcinoma"

_biomedicines, 2023, doi:10.3390/biomedicines11061528_

Round 1
Reviewer 1 Report
The authors well summarized and digested recent information about molecular mechanisms related to nasopharyngeal carcinoma (NPC), which include candidates of novel target for NPC treatments. This compact review, which involves comprehensive reports, is giving the head and neck oncologists some new hints of future research about NPC. Some following issues should be reconsidered.
In Table 2, about EBNA1, What does PML stand for?
Line 237, western blotting had better be Western blotting.
Line 266, 2014 paper?
Line 303, TGF-b. Beta?
Line 350-351, These sentences need a reference.
Line 352-353, "EBV-driven global-wild epigenetic silencing of many tumor suppressor genes." This lacks a verb. Is it intentional?
This is a refined review for understanding NPC.
Author Response
Reviewer 1
Comments and Suggestions for Authors
The authors well summarized and digested recent information about molecular mechanisms related to nasopharyngeal carcinoma (NPC), which include candidates of novel target for NPC treatments. This compact review, which involves comprehensive reports, is giving the head and neck oncologists some new hints of future research about NPC. Some following issues should be reconsidered.
Response
Thank you very much for your variable suggestions to improve our manuscript. We remark the corrected parts in red colors. And we prepare a point-to-point response according to your suggestions:
In Table 2, about EBNA1, What does PML stand for?
Response:
Thank you very much for your suggestions. We added the abbreviation to PML as “Abbreviation: PML, Promyelocytic leukemia”. ( line 125)
Line 237, western blotting had better be Western blotting.
Response:
Thank you very much for your suggestions. We revise western blotting into Western blotting as “By EMSA, Western blotting, and immunohistochemical staining, constitutive activa-tion of NF-κB complexes, either p50/p50/Bcl3 or p50/RelB, was found in almost all EBV-positive NPC tumors and cell lines”(line 241)
Line 266, 2014 paper?
Response:
Thank you very much for your suggestions.
We added the reference at ” In an NPC GWAS, genes involved in the ErbB-PI3K signaling pathway were mutated in 30/128 cases[53]”.(line 269-270)
Line 303, TGF-b. Beta?
Response:
Thank you very much for your suggestions.
We revise TGF-b. into TGF-β as” TGF-β signaling regulates G1/S cell cycle transition, apoptosis, senescence, epitheli-al-to-mesenchymal transition (EMT), metastasis, angiogenesis and immune responses [88]”(line 307)
Line 350-351, These sentences need a reference.
Thank you very much for your suggestions.
We added the reference at” One study found that ARID1A mutations were present in 17.5% of NPC cases and were associated with poorer overall survival[95].”(line 354-355)
Line 352-353, "EBV-driven global-wild epigenetic silencing of many tumor suppressor genes." This lacks a verb. Is it intentional?
Thank you very much for your suggestion and remind.
We revised this paragraph as” EBV infection plays a critical role in epigenetic alteration. EBV-driven global-wild epigenetic silencing includes many tumor suppressor genes. Key epigenetic drivers for (1) DNA methylation (DNMT1, TET1, IDH2), (2) histone modification (KMT2c, EZH2, EED, and BMI1), and chromatin remodeling (ARID1A, BRD7, PRDM5, EP300 and HELLS) were identified [2].”

Reviewer 2 Report
Overall, it is a well-organized and updated overlook on NPC tumorigenesis. Some parts are quite scholastic though and others deserve more discussion:
- I would add a separate paragraph + figure where the embryology/anatomy/histology of NP are briefly discussed: the importance of the MALT system and the interplays between the respiratory epithelium and the immune system starts from these very aspects (for instance, 10.1038/mi.2010.1016 or /B978-0-12-811924-2.00002-X or https://www.ncbi.nlm.nih.gov/pmc/articles/PMC7097243/ )
- please discuss in more detail histopatologic analysis of NPC and the role of molecular biology versus IHC for the identification of EBV in biopsies (you can also provide a panel of figures with NP appearance by endoscopy, microscopic appearance etc.)
- early diagnosis: discuss the role of nasoendoscopy and new bioimaging techniques (NBI)
- Figure 1 is poorly presented: what kind of epithelial cells are represented? where is the lamina propria? EBV DNA increases in the microenvironment or in the blood? make it clearer since good figures are essential for this kind of manuscripts.
Author Response
Reviewer 2
Comments and Suggestions for Authors
Overall, it is a well-organized and updated overlook on NPC tumorigenesis. Some parts are quite scholastic though and others deserve more discussion:
Response:
Thank you very much for your variable suggestions to improve our manuscript. We remark the corrected parts in red colors. And we prepare a point-to-point response according to your suggestions:
- I would add a separate paragraph + figure where the embryology/anatomy/histology of NP are briefly discussed: the importance of the MALT system and the interplays between the respiratory epithelium and the immune system starts from these very aspects (for instance, 10.1038/mi.2010.1016 or /B978-0-12-811924-2.00002-X or https://www.ncbi.nlm.nih.gov/pmc/articles/PMC7097243/ )
Response:
Thank you very much for your suggestions. We added one paragraph to discuss the importance of the MALT system as “Nasopharynx-associated lymphoid tissue (NALT) is different from other lymphoid tissue from organogenesis. NALT plays an important role in mucosa immune response [96]. Secretion of EBV-specific IgA from NALT to the serum can serve as a tool for NPC screening in high-risk groups.” (line 362-365 )
- please discuss in more detail histopatologic analysis of NPC and the role of molecular biology versus IHC for the identification of EBV in biopsies (you can also provide a panel of figures with NP appearance by endoscopy, microscopic appearance etc.)
Response:
Thank for your suggestion, again; we added one paragraph as “Histologically, the NPC is classified into WHO type I, II and III according to keratinizing and differentiation. Molecular markers, including genetic factors and EBV variants, are potentially useful in subtyping NPC for prognosis and treatment [2].” (line 154-157)
Because this review is focused on the molecular mechanism of NPC, we did not add further figures with NP appearance by endoscopy or microscopic appearance.
- early diagnosis: discuss the role of nasoendoscopy and new bioimaging techniques (NBI)
Response:
We discuss the role of nasoendoscopy in the additional paragraph with the statement: “ Nasopharyngoscopy is the tool of choice to exam nasopharynx and guide biopsy. Narrow-band imaging (NBI) could be applied for the diagnostic screening and accuracy rate increment of a high-risk population of NPC [19].” (line 100-102)
- Figure 1 is poorly presented: what kind of epithelial cells are represented? where is the lamina propria? EBV DNA increases in the microenvironment or in the blood? make it clearer since good figures are essential for this kind of manuscripts.
Response:
Thank for your suggestions. We have revised Figure 1 and marked epithelial cells, lamina propria, and plasma EBV DNA.

Round 2
Reviewer 2 Report
Thanks for accepting my suggestions